# Fungal Diversity and Its Relationship with Environmental Factors in Coastal Sediments from Guangdong, China

**DOI:** 10.3390/jof9010101

**Published:** 2023-01-11

**Authors:** Keyue Wu, Yongchun Liu, Xinyu Liao, Xinyue Yang, Zihui Chen, Li Mo, Saiyi Zhong, Xiaoyong Zhang

**Affiliations:** 1University Joint Laboratory of Guangdong Province, Hong Kong and Macao Region on Marine Bioresource Conservation and Exploitation, College of Marine Sciences, South China Agricultural University, Guangzhou 510642, China; 2Guangdong Provincial Key Laboratory of Aquatic Product Processing and Safety, College of Food Science and Technology, Guangdong Ocean University, Zhanjiang 524088, China; 3Guangdong Laboratory for Lingnan Modern Agriculture, Guangzhou 510642, China

**Keywords:** coastal sediments, marine fungi, environmental factors, diversity, composition, trophic types

## Abstract

As one core of the Guangdong–Hong Kong–Macao Greater Bay Area (GBA), Guangdong is facing some serious coastal environmental problems. Fungi are more vulnerable to changes in coastal environments than bacteria and archaea. This study investigated the fungal diversity and composition by high-throughput sequencing and detected basic parameters of seven environmental factors (temperature, dissolved oxygen, pH, salinity, total organic carbon, total nitrogen, and total phosphorus) at 11 sites. A total of 2056 fungal operational taxonomic units (OTUs) belonging to 147 genera in 6 phyla were recovered; *Archaeorhizomyces* (17.5%) and *Aspergillus* (14.19%) were the most dominant genera. Interestingly, a total of 14 genera represented the first reports of coastal fungi in this study. Furthermore, there were nine genera of fungi that were significantly correlated with environmental factors. FUNGuild analysis indicated that saprotrophs and pathogens were the two trophic types with the highest proportions. Saprotrophs were significantly correlated with total organic carbon (TOC), total nitrogen (TN), and total phosphorus (TP), while pathogens were significantly correlated with pH. This study provides new scientific data for the study of the diversity and composition of fungal communities in coastal ecosystems.

## 1. Introduction

As the interface between land and sea, the coastal region not only provides many natural resources and facilities for human beings but also has a significant impact on nutrient cycling, decomposition, biodiversity conservation, and pollutant degradation in the ocean [1]. Moreover, the coastal region accumulates nutrients, heavy metal pollutants, and organic pollutants, and there are frequent changes in the coastal environment (such as salinity, dissolved oxygen, and nutrient content) caused by tidal movement, resulting in a much greater diversity of microbial communities in coastal sediments than in other sediments [2].

Although there have been many studies on the microbial communities in coastal environments, most studies have focused on the changes in bacterial communities’ structural diversity in sediments under different environmental conditions [3]. In recent years, fungi derived from marine environments have attracted increasing attention because of their significant ecological roles, and some studies have revealed that fungi are more vulnerable to or affected by their environments than bacteria and archaea [4,5,6].

Fungi can contribute to the decomposition of organic matter, nitrogen fixation, and phosphorus dissolution, thereby promoting plant growth [7]. Many studies have confirmed that changes in fungal community composition are related to environmental factors such as salinity [8], dissolved oxygen [9], and the pollution of marine sediments [10]. Moreover, human activities can cause changes in environmental factors, such as the release and transport of pollutants into the environment [7,11]. Recent studies have demonstrated that environmental changes influence the functional composition of fungi more precisely than their taxonomic composition [12], and changes in microbial function may be a more sensitive signal of environmental change.

The Guangdong–Hong Kong–Macao Greater Bay Area (GBA) is a region of rapid growth and economic vitality. As a key geographic node for land, sea, and air transport, the GBA is significantly involved in building the Overland Silk Road Economic Belt and the important Silk Road of the 21st Century [13]. In the past 20 years of the GBA’s development, rapid economic growth has brought about a series of ecological and environmental problems, and the GBA is also facing increasing pressure on resources and the environment. Therefore, this study selected some sites in the coastal areas of Guangdong Province and the GBA to study their environmental physicochemical properties and fungal diversity, providing a reference for ecological research in this region.

The purpose of this study was to investigate the correlation between fungi and environmental factors in sediments at 11 sites in Guangdong, China. To the best of our knowledge, this is the first detailed report on the coastal fungal diversity in Guangdong Province, providing new scientific data for the study of coastal fungal diversity as well as a new scientific basis for follow-up studies on the correlation between coastal fungal communities and environmental factors.

## 2. Materials and Methods

### 2.1. Study Sites and Sample Collection

According to the degree of impact of social and economic activities, typical sediment sites were selected in the coastal areas of Guangdong Province as research objects (Figure 1), and sediment samples were collected in December 2019. Starting from the Shanwei area (S1), the sampling sites stretched across eight coastal cities in Guangdong Province from east to west along the coastline (six stations, from S3 to S8, were in the GBA) and ended in the Zhanjiang area (S11). A set of sterile stainless steel grips were used to collect samples from different sites in 3–5 cm of surface sediment at each site, which was placed in sterile glass containers. All samples were collected at low tide, and the tidal flats were submerged and exposed approximately twice a day. Three replicates (e.g., S1a, S1b, S1c) were collected (at a 0.5 m horizontal interval) from each site, and a total of 33 sediments were collected from 11 sites. The collected sediments were mainly composed of sand, and the impurities (such as shellfish and plant roots) were removed during sediment collection. Then, the collected sediments were placed in ice and transferred to storage at −80 °C as soon as possible for subsequent experiments. All sediment samples were divided into two portions, for the examination of environmental factors and fungal diversity, respectively.

### 2.2. Determination of Environmental Factors

A multifunction water quality analyzer (YSI 556MPS, Yellow Springs, OH, USA) was used to measure the temperature (T), pH, dissolved oxygen (DO), and salinity of the coastal water [14]. The freeze-dried sediment samples were sieved through a 200-mesh screen (~74 μm) after homogenization and grinding. The screened samples were divided into two parts: one was pretreated with 1 M HCl for the determination of total organic carbon (TOC), while another (untreated) sample was used for the determination of total nitrogen (TN) [15]. TOC and TN were tested by using the CHNOS Elemental Analyzer (Vario EL Cube, Langenselbold, Germany) [16]. The relative error for TOC and TN was 6% and 1.2%, respectively, based on duplicate measurements [17]. The total phosphorus (TP) in the sediment was extracted with 1 M HCl after combustion for 2 h at 500 °C [18]. The ascorbic acid–molybdate blue method was used to test the amount of total phosphorus (TP) in the sediment samples [19].

### 2.3. DNA Extraction and Sequencing Analysis of Fungi

The FastDNA^®^ Spin Kit was used to recover total genomic DNA from the sediment samples (MP Biomedicals, Santa Ana, CA, USA). The ITS sequences in the fungal DNA were amplified by polymerase chain reaction (PCR) using the universal primers ITS3F (5′-GCATCGATGAAGAACGCAGC-3′) and ITS4R (5′-TCCTCCGCTTATTGATATGC-3′) [20]. PCRs were conducted using a 50 μL mixture containing 25 μL of Master Mix (New England Biolabs, Ipswich, MA, USA), 2 μL of each primer, 2 μL of DNA (~20 ng/μL), and 21 μL of nuclease-free water. The following PCR conditions were used: 95 °C for 5 min, 30 cycles at 94 °C for 30 s, annealing at 54 °C for 30 s, extension at 72 °C for 30 s, and 72 °C for 7 min. All PCR products from each fungal sample were purified by using an Agencourt AMPure XP Kit (Beckman Coulter, Brea, CA, USA). The ITS rDNA sequencing library was constructed, and the validated DNA library concentration was detected using a Qubit 3.0 fluorometer. Finally, the fungal library was sequenced on the Illumina MiSeq platform (Illumina, San Diego, CA, USA) [21].

To obtain high-quality and clean reads, Quantitative Insights Into Microbial Ecology (QIIME) (http://qiime.org/, accessed on 1 March 2020) was used for qualitative filtering of the raw sequences. FLASH was used to merge the two terminal sequences [22]. Then, the sequences were divided into operational taxonomic units (OTUs) based on a 97% similarity threshold value by using the clustering program VSEARCH (v1.9.6) [23]. The UNITE database (version v8_02.02.2019) [24] and the International Mycological Association web (www.mycobank.org, accessed on 23 December 2022) were used to identify the taxonomic diversity of the fungal communities.

### 2.4. Statistical Analyses

The means and standard deviations of environmental factors were calculated using RStudio (Version 1.4.1717). The alpha diversity of the fungal communities was evaluated with the Shannon, Chao1, Simpson, and ACE indices using the “vegan” package in RStudio. The beta diversity of the fungal communities was evaluated via principal coordinate analysis (PCoA) using the “ape”, “vegan”, and “ggplot2” packages in RStudio. The significance of environmental factors and alpha diversity was determined by one-way ANOVA analysis (LSD, *p* < 0.05) using SPSS v26.0. The FUNGuild database (http://www.stbates.org/guilds/app.php, accessed on 1 May 2020) was used for fungal functional prediction. Pearson’s correlation analysis (PCA) was used to analyze the correlation of the fungal genera and trophic types with environmental factors in RStudio.

### 2.5. Accession Numbers for Nucleotide Sequences

All sequence data from coastal sediments were deposited in the Sequence Read Archive of the NCBI under accession number PRJNA833334.

## 3. Results

### 3.1. Physicochemical Properties of Coastal Sediments

The latitudes and longitudes of coastal sediments at 11 sites along Guangdong are shown in Appendix A. The physicochemical properties of the coastal sediments from the 11 sites along Guangdong are shown in Appendix A. The temperature fluctuated within a range of 21.3~24.5 °C and pH within a narrow range of 8.02 to 8.77. The concentration of dissolved oxygen (DO) fluctuated greatly at different sites, with DO concentrations as high as 8.24 mg/L at site S1 and as low as 4.43 mg/L at site S6. The salinity of the coastal water also varied greatly. The salinity at sites S6, S11, and S5 was less than 16‰ (4.56‰, 10.50‰, and 14.65‰, respectively), while the salinity at the remaining sites ranged from 24.34‰ to 33.23‰. The total organic carbon (TOC) concentration in the sediments at different sites ranged from 546.48 to 3596.32 mg/kg, total nitrogen (TN) was between 101.20 and 342.72 mg/kg, and total phosphorus (TP) was between 175.61 and 1980.49 mg/kg.

### 3.2. Composition and Distribution of Fungi in Coastal Sediments

The results of the analysis showed that 2,166,705 effective fungal ITS sequences were obtained and clustered into 2056 OTUs with 97% sequence similarity (Appendix A). A plateau was seen in all 11 rarefaction curves, and the Good’s coverage values were 0.99 at all sites, suggesting that all 11 sites might be adequately represented by the ITS sequences observed. The Chao1, ACE, Simpson, and Shannon indices (Table 1) indicated the consistency of the abundance and diversity of fungi in the 11 coastal sediments. The abundance and diversity of fungi at sites S1 and S3 were higher, and the abundance and diversity of fungi at site S5 were lower than those at the other sites.

Among the 2056 OTUs of the fungi, there were 1379 OTUs that remained unclassified at the phylum level, and the remaining 677 OTUs belonged to 147 genera in 6 phyla. Ascomycota (80.02%) was the predominant phylum of fungi, followed by Basidiomycota (14.86%), Chytridiomycota (2.98%), Mortierellomycota (0.61%), Glomeromycota (0.21%), and Neocallimastigomycota (0.17%) (Appendix A) [25]. For the classified genera (Figure 2A), *Archaeorhizomyces* (16.21%), *Aspergillus* (13.05%), *Didymosphaeria* (7.22%), *Fusarium* (6.04%), *Laetisaria* (4.43%), *Phoma* (4.41%), *Geomyces* (3.40%), *Massariosphaeria* (3.32%), *Candida* (2.57%), *Spirosphaera* (2.13%), *Pichia* (1.72%), *Saccharomyces* (1.39%), *Nakaseomyces* (1.17%), *Pseudogymnoascus* (1.12%), *Wardomyces* (1.10%), *Malassezia* (1.03%), *Cladosporium* (0.99%), *Camarophyllopsis* (0.76%)m and *Mortierella* (0.72%) were the top 19 genera, and approximately 13.07% of the sequences belonged to unclassified fungi at the genus level.

In particular, through relevant research of 147 genera of fungi, it was found that 14 genera of fungi were detected for the first time in the coastal environment, among which the abundance of 2 genera—*Laetisaria* [26] and *Spirosphaera* [27]—ranked in the top 10 of all fungi. The remaining 12 fungal genera were as follows: *Gibellulopsis* [28], *Monacrosporium* [29], *Spegazzinia* [30], *Rhizopycnis* [31], *Dokmaia* [32], *Vermispora* [33], *Hirsutella* [34], *Erythricium* [35], *Limonomyces* [36], *Entorrhiza* [37], *Dinemasporium* [38], and *Villosiclava* [39].

The 19 most abundant fungal genera were observed at almost every site (Figure 2B), but their proportions were different. For example, *Archaeorhizomyces* and *Aspergillus* existed at all 11 sites, and they were dominant fungi at S1, S2, S3, S5, S8, S9, S10, and S11. *Laetisaria* was the dominant genus at S4 but did not exist at S2, S3, S5, or S8. *Cladosporium* existed only at S1, S5, S6, and S9.

Principal coordinate analysis (PCoA) was used to analyze the composition of the fungal communities at different locations. The results (Appendix A) showed that the subsamples at S2, S3, S7, and S11 were discrete, while the subsamples at S1, S4, S9, and S10 showed significant aggregation. The S5 and S8 subsamples showed obvious aggregation, while the overall sites of S3 and S4 were far away from other locations. Samples from site S1 to site S4 were relatively independent, while some molecular samples from site S5 to S11 were aggregated.

### 3.3. Functional Prediction of Fungi by FUNGuild

Functional prediction of fungi was performed using the FUNGuild database, and they were analyzed in pairs with locations and environmental factors (Figure 3). FUNGuild was used to classify fungi as undefined saprotrophs (27.70%), plant pathogens (15.37%), endophytes (10.77%), lichen parasites (9,14%), animal pathogens (8.15%), wood saprotrophs (7.44%), soil saprotrophs (6.93%), dung saprotrophs (3.67%), litter saprotrophs (2.94%), ectomycorrhizal (2.57%), animal endosymbionts (1.05%), fungal parasites (1.03%), leaf saprotrophs (0.84%), bryophyte parasites (0.84%), lichenized (0.48%), plant saprotrophs (0.45%), epiphytes (0.43%), arbuscular mycorrhizal (0.20%), and ericoid mycorrhizal (0.01%) fungi (Figure 3A). However, there was a large difference in the proportion of each trophic mode at each site (Figure 3B). For example, the proportion of plant pathogens at site S4 was higher than that at the other sites, while the proportions of animal pathogens, wood saprotrophs, and soil saprotrophs at site S6 were higher than those at the other sites.

### 3.4. Correlation Analysis between Fungi and Environmental Factors

Redundancy analysis (RDA) was conducted on the horizontal community structure of OTUs and seven environmental factors of fungi in the sediments (Figure 4). The explanatory degrees of the first and second ordination axes were 50.71% and 31.79%, respectively. Salinity was positively correlated with the first axis (*p* = 0.03), while DO was negatively correlated with the second axis (*p* = 0.01). DO and salinity were the most important factors affecting the fungal communities.

It was found that among the top 19 fungal genera using PCA, the fungal composition of 9 genera was significantly correlated with environmental factors (Figure 5A). These genera were *Massariosphaeria*, *Saccharomyces*, *Aspergillus*, *Didymosphaeria*, *Pichia*, *Cladosporium*, *Laetisaria*, *Fusarium*, and *Phoma*. Among them, *Phoma* was negatively correlated with T. The genera *Laetisaria*, *Fusarium*, and *Phoma* were significantly negatively correlated with pH, of which *Laetisaria* and *Phoma* were extremely significant. *Aspergillus* and *Didymosphaeria* were positively correlated with DO. *Cladosporium*, *Fusarium*, and *Phoma* were negatively correlated with salinity, of which *Phoma* was extremely significant. *Massariosphaeria* and *Cladosporium* were significantly correlated with TOC; *Massariosphaeria* had a significant negative correlation with TOC, while *Cladosporium* had a significant positive correlation with TOC. *Saccharomyces* and *Pichia* showed a significant correlation with TN; *Saccharomyces* was significantly negatively correlated with TN, while *Pichia* had a significant positive correlation with TN. *Aspergillus* was significantly negatively correlated with TP.

### 3.5. Correlation Analysis between Trophic Modes and Environmental Factors

According to a heatmap (Figure 5B), 7 environmental factors and 11 trophic modes (fungal parasite, leaf saprotroph, bryophyte parasite, epiphyte, ericoid mycorrhizal, plant pathogen, soil saprotroph, animal pathogen, wood saprotroph, and dung saprotroph) were significantly correlated. Among them, wood saprotrophs and dung saprotrophs showed a significant negative correlation with T. Eight trophic modes (fungal parasite, leaf saprotroph, bryophyte parasite, epiphyte, plant pathogen, soil saprotroph, litter saprotroph, and animal pathogen fungi) had a significant correlation with pH. There was a significant positive correlation between fungal parasites and pH. Leaf saprotrophs and bryophyte parasites had an extremely significant positive correlation with pH. Epiphytes, plant pathogens, soil saprotrophs, litter saprotrophs, and animal pathogens showed a significant negative correlation with pH. Four types of trophic modes (fungal parasites, leaf saprotrophs, bryophyte parasites, and ericoid mycorrhizae) were significantly correlated with DO, among which fungal parasites, leaf saprotrophs, and bryophyte parasites were significantly positively correlated with DO, while ericoid mycorrhizae were significantly negatively correlated with DO. Six trophic modes (ericoid mycorrhizal, soil saprotroph, litter saprotroph, animal pathogen, wood saprotroph, and dung saprotroph fungi) had a negative correlation with salinity, and soil saprotrophs and litter saprotrophs had a significant negative correlation with salinity. Ericoid mycorrhizae were positively correlated with TN.

## 4. Discussion

Coastal sediments are exposed to extreme marine environmental conditions that reflect the conditions of seawater and coastal areas, in which biomass is very important. This research sought to study the effects of environmental factors on the diversity and composition of fungi in sediments.

### 4.1. Environmental Factors in Coastal Sediments

By measuring the environmental factors of coastal sediments in Guangdong Province, it was found that there were some similarities and differences in the environmental factors of different sites (Appendix A). The temperature difference of the 11 sites was 3.2 °C, which might have been related to local weather conditions. Site S1 had the highest DO, which might have been caused by clear weather in a certain period before sampling. At the same time, this spot was also a scenic spot with a good natural environment. DO was the lowest at S4, which might have been due to cloudy and rainy weather, low air pressure, and decreased dissolved oxygen in the water before sampling. The salinity at S6 was significantly lower than that at the other sites because it was fed by rivers. Compared with previous studies, the pH value of coastal water in Guangdong Province was similar to that of coastal water in the Bohai Sea and Yellow Sea, while the DO and salinity at most sites were higher than those of coastal water in the Bohai Sea and Yellow Sea [40].

C, N, and P are important nutrients, and various forms of C, N, and P are very important for assessing the response of aquatic ecosystems to environmental changes and the impact of human activities [41,42]. In the investigation and statistics of this study, the TOC and TN at sites S3–S8 located in the Guangdong–Hong Kong–Macao Greater Bay Area almost all showed high levels, especially at sites S5 and S6. Site S5 was in the Zhuhai Chimelong International Ocean Resort, which might cause the discharge of domestic water, resulting in high TOC and TN on the nearby coast. Site S6 was on Zhuhai’s Qi‘ao Island. Facing Hong Kong and Shenzhen in the east, Qi‘ao Island is in the center of the Golden Triangle of Guangzhou, Hong Kong, and Macao. It was the only way for the construction of the Lingdingyang Sea Bridge. The high levels of TOC and TN might have also been due to frequent human activities [7,40]. The contents of total phosphorus at sites S2, S3, S5, and S6 were all at high levels, which might have been caused by human activities and excessive fertilization [43].

### 4.2. Fungal Diversity and Distribution

According to the Chao1, ACE, Simpson, and Shannon indices of fungal communities at the 11 sites in coastal areas, the richness and diversity of the fungi at each site tended to be consistent. Compared with previous studies on fungal diversity in the coastal sediments of the East China Sea [44] and the mangrove sediments of the Maowei Sea in Guangxi [7], it was found that the fungal diversity in the East China Sea and the mangrove sediments of Guangxi was higher than that in the coastal areas of Guangdong, which might be related to the species diversity caused by different environments.

A total of 147 genera in 6 phyla were found in the investigation of fungal diversity in the coastal sediments of Guangdong. Ascomycota accounted for 80.02% at the phylum level, followed by Basidiomycota (14.86%) and Chytridiomycota (2.98%). These phyla have been found in coastal areas by many researchers [4,7,45]. For example, Lin et al. [46] also found that the most dominant fungi were Ascomycota and Basidiomycota in a study of fungal diversity in typical coastal sediments in Chinese sea areas. This showed that the fluctuation of the horizontal structure of the phyla was small in terms of spatial position. Further subdivided at the taxonomic level, according to an analysis of the relative abundance of the top 19 genera (Figure 2), the dominant fungal genera were *Archaeorhizomyces* (16.21%) and *Aspergillus* (14.14%). This conclusion was not surprising, as these two genera of fungi are widely distributed in a variety of ecosystems around the world [47]. In terms of trophic types, they are potential saprotrophic microorganisms. For example, *Archaeorhizomyces* are often found in plant roots and rely on the organic matter provided by roots [48]. This was also consistent with the results of the functional analysis.

In this study, 14 genera of different fungi were found to exist in coastal environments for the first time. The discovery of these fungi for the first time in a coastal environment demonstrates the diversity and complexity of fungi. There is still a long way to go to explore the factors affecting fungal communities. This study also provides new scientific data for studying the diversity and composition of fungal communities.

The diversity and composition of the fungi varied between the different sites. According to the PCoA results of the 11 sites, the fungal community composition of sites S1–S4 was significantly different from that of the other sites, and the fungal community composition of sites S5–S11 was relatively similar. The separation between S4 and S5 by the Pearl River Estuary might be the main reason for this difference in fungal communities. This also indicates that there are differences in fungal community structure between the eastern and western coastal areas of Guangdong Province. In addition to the differences in fungal structure caused by the environmental factors studied in this paper, population growth and industrial expansion [49] are also factors that disturb the coastal environment and cause differences in fungal diversity.

### 4.3. Correlation Analysis between the Fungal Communities and Environmental Factors

The RDA plots showed that DO and salinity were the most important factors affecting the fungal communities. The research of Wang et al. [23] showed that salinity was an important factor affecting the β-diversity of coastal fungal communities. The study by Cathrine and Raghukumar [50] also confirmed that DO was an important factor affecting fungal communities in coastal sediments, and they hypothesized that DO and salinity might affect fungal communities through denitrification and ammonia nitrogen. In addition, DO was negatively correlated with salinity, which is consistent with the findings of Barik et al. [51].

PCA was used to analyze the correlation between the top 19 fungal genera and environmental factors. In this study, nine genera of fungi showed significant correlations with environmental factors. *Massariosphaeria*, which was less well reported and was known to have been isolated from fresh water [27], was negatively correlated with TOC. *Saccharomyces* and *Pichia* showed a significant negative correlation with TN. Interestingly, these are both common yeasts that are used in commercial fermentation [52]. According to previous reports on the effects of nitrogen content on *Saccharomyces* and *Pichia* fermentation [53,54], *Saccharomyces* and *Pichia* showed a negative correlation with TN, which might have been due to their preference for nitrogen sources or C/N ratio differences [55]. *Aspergillus* showed a significant negative correlation with TP, and Tian et al. [56] also reported similar results. They indicated that the increases in C, N, and P could lead to an increase in the abundance of *Cladosporium* and a decrease in the abundance of *Massariosphaeria*, *Saccharomyces*, and *Aspergillus*.

*Didymosphaeria* and *Aspergillus*, which were the second and third most abundant genera, were significantly affected by DO, which might be one of the reasons for the impact of DO on fungal communities. Previous studies reported that *Fusarium* and *Phoma* had high abundance in low-salinity waters, which is consistent with the results of this study [57,58]. In this paper, the conclusion regarding *Cladosporium* and salinity was inconsistent with the findings of previous studies, suggesting that the reason for this relationship might also be related to the joint action of the dominant fungal community or multiple environmental factors [59].

### 4.4. Correlation Analysis between Fungal Function Prediction and Environmental Factors

The FUNGuild functional classification predicted that saprotrophs were the dominant trophic type in the coastal sediments of Guangdong Province. Saprotrophs have been identified as the most abundant nutrient mode for fungi in a variety of environments, including coastal sediments [60]. Among the nine genera that were significantly correlated with environmental factors, eight genera were saprotrophs or trophic types that include saprotrophs, and all belonged to Ascomycota. The study by Wang et al. [61] also showed that community changes in saprotrophs and fungi from Ascomycota were affected by TOC, TN, and TP. Saprotrophic fungi were more strongly driven by nutrient factors than other nutrient types [62]. In addition, saprotrophs are important decomposers in the ecosystem and consume nutrients by degrading apoptotic cells [63]. Moreover, various types of saprotrophs were significantly correlated with T, pH, DO, and salinity. These results indicated that saprotrophs were indicators of the ecological environment and played an essential role in the ecosystem.

In this study, pathogens were in second place with regard to the trophic types of fungi. Pathogenic fungi often harm their hosts and threaten the ecological environment [64]. Among the nine genera of fungi that were significantly correlated with environmental factors, five were pathogens or trophic types containing pathogens. As a representative genus of plant pathogens, *Fusarium* was one of the most abundant fungi in both coastal and deep-sea sediments [65,66], which might be attributed to the presence of microscopic plant fragments in the sediments [67]. The animal pathogen *Pseudogymnoascus*, as the dominant species found in marine environments [68], has been found to have the ability to become saprotroph in hibernaculum environments [69]. This suggests that humus in sediments is also a possible source of pathogens. In addition, changes in natural conditions are also a factor that can lead to an increase in pathogens, such as precipitation [70]. In this research, the abundance of both plant pathogens and animal pathogens increased significantly with decreasing pH. Comprehensive correlation analysis suggested that pathogenic fungi in the sediments mainly had a strong correlation with pH. Although a study by Su et al. [11] showed that acidification only increased the abundance of animal pathogens in sediments, the abundance of plant pathogens was also significantly higher in our results. Animal pathogens may pose a potential threat to aquaculture and human safety by interacting with organisms higher up the food chain and becoming parasitic in fish, shrimp, and shellfish [71]. Therefore, our results shed light on the potential relationships between pathogens and environmental factors and improve the understanding of the environment and human health.

As a tool that connects fungal gene sequences with nutrient types, the FUNGuild database provides a good foundation for studying the ecological functions of fungi. Lin et al. used the FUNGuild database to predict the functions of fungi in different fertilizers in the Loess Plateau, finding that functional groups were more sensitive to fertility than community composition [72]. Li et al. predicted the functions of soil fungi in tea orchards and found that long-term cultivation of tea trees would enrich saprotrophs and plant pathogens, reducing the abundance of beneficial fungi [73]. The FUNGuild database is of great help in studying the relationships between fungi and ecology, but it has its own limitations that cannot be ignored. Yan et al. found that *Aspergillus candius * was classified as a saprotroph in the FUNGuild database, while their report had classified it as an endophytic fungus in plants [74,75]. The data of the FUNGuild database are not updated in real time, and a large amount of fungal information has not been collected. Therefore, it can be combined with other databases, such as FungalTraits [76] and FacesOfFungi [77] to analyze the functions of fungi.

## 5. Conclusions

In this study, the levels of seven environmental factors (T, pH, DO, salinity, TOC, TN, and TP) were different at 11 coastal sediment sites in Guangdong Province, and pH and salinity were significantly correlated with some pathogens (*Cladosporium*, *Phoma*, *Fusarium*, etc.). These results also indicate that environmental factors might influence the ecological environment of coastal sediments by influencing the fungal community structure. At the same time, our study reported for the first time that 14 genera, including *Laetisaria*, were detected in coastal regions. To the best of our knowledge, this is the first report of the correlation of environmental factors at different sites along the Guangdong coast with fungal communities and nutrient types in sediments. This provides a new scientific basis for the study of the coastal ecological environment in Guangdong Province and the relationships between fungal diversity and environmental factors. However, due to the diversity of environmental factors and database limitations, many fungal OTUs were not classified. The incomplete information of the FUNGuild database and the relationships between fungal communities in coastal sediments and environmental factors still need further exploration and consideration.

## Figures and Tables

**Figure 1 jof-09-00101-f001:**
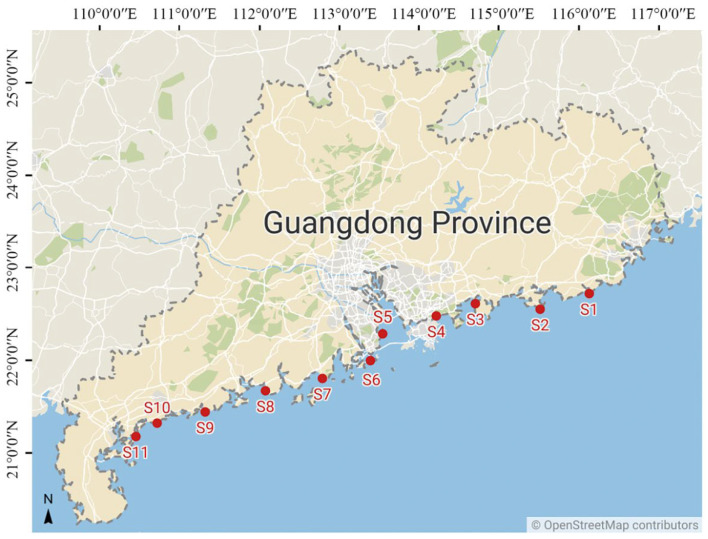
Location of 11 sampling sites along the Chinese coastal region.

**Figure 2 jof-09-00101-f002:**
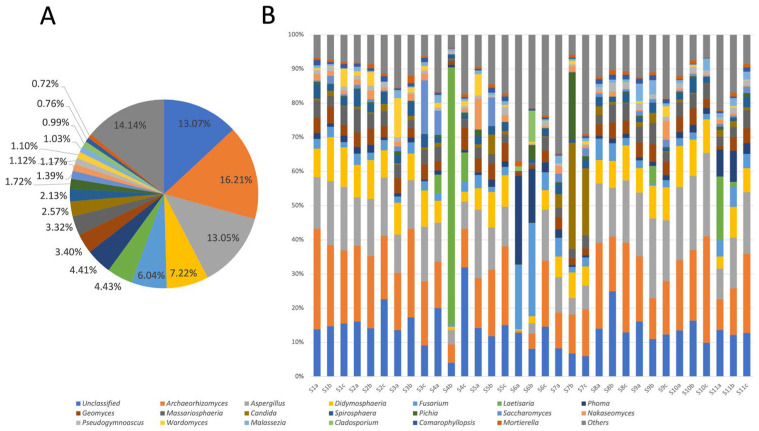
The (**A**) fungal composition and (**B**) genus distribution of 33 samples from 11 coastal locations (S1–S11) in Guangdong, South China. Three replicates (e.g., S1a, S1b, S1c) were collected from each site.

**Figure 3 jof-09-00101-f003:**
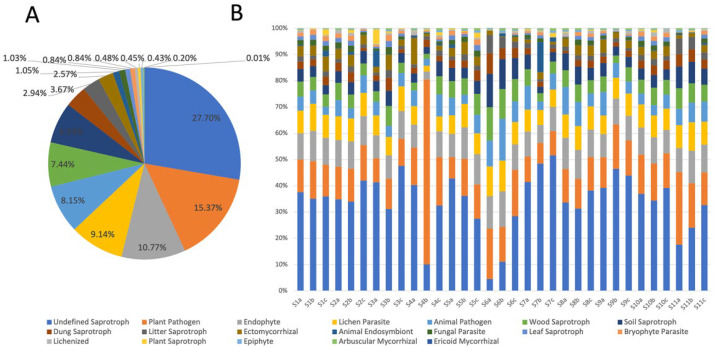
The (**A**) composition and (**B**) distribution of fungal trophic types in coastal sediments at 11 sites (S1–S11) in Guangdong province. Three replicates (e.g., S1a, S1b, S1c) were collected from each site.

**Figure 4 jof-09-00101-f004:**
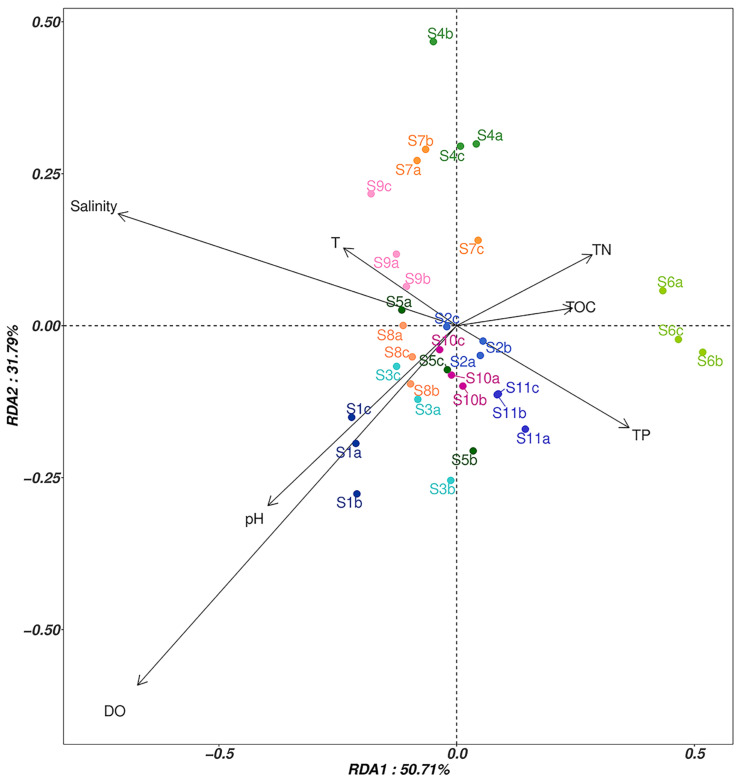
RDA analysis of fungal community structure and environmental factors at 11 coastal sites (S1–S11). Three replicates (e.g., S1a, S1b, S1c) were collected from each site.

**Figure 5 jof-09-00101-f005:**
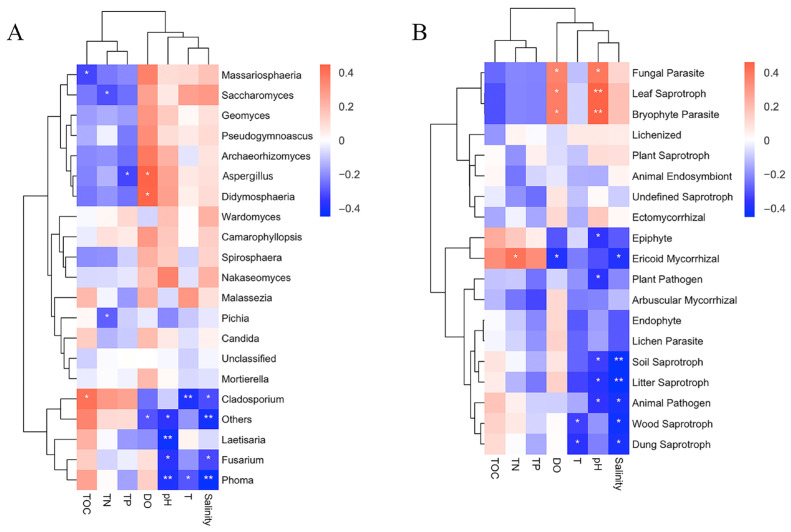
Heatmap of the correlation between (**A**) fungal community composition (top 20) and environmental factors of taxonomic genera, and (**B**) fungal trophic modes composition (top 19) and environmental factors.

**Table 1 jof-09-00101-t001:** Comparison of fungal sequences, the operational taxonomic unit (OTU) richness, and diversity indices of the ITS sequences for clustering at 97% similarity from the sequencing analysis.

Sites	Sequence Read	OTUs	Chao1	ACE	Shannon	Simpson
S1	83,567 ± 4507 ab	151 ± 4 c	486 ± 50 ab	466 ± 39 ab	5.8 ± 0.55 ab	0.91 ± 0.05 ab
S2	78,526 ± 9626 ab	150 ± 3 c	444 ± 72 ab	434 ± 70 ab	4.6 ± 1.40 b	0.80 ± 0.18 ab
S3	70,227 ± 8075 b	179 ± 1 a	514 ± 35 a	495 ± 27 a	6.1 ± 0.27 a	0.95 ± 0.01 a
S4	66,106 ± 5547 b	156 ± 5 bc	423 ± 42 b	418 ± 44 b	6.0 ± 0.41 a	0.94 ± 0.02 ab
S5	73,033 ± 9366 ab	154 ± 4 bc	319 ± 14 c	296 ± 14 c	4.0 ± 0.36 b	0.69 ± 0.06 b
S6	86,543 ± 16,540 a	181 ± 21 a	369 ± 65 bc	366 ± 66 bc	5.2 ± 1.07 ab	0.90 ± 0.08 ab
S7	86,324 ± 12,343 ab	158 ± 6 bc	421 ± 50 b	411 ± 50 b	5.0 ± 0.34 ab	0.86 ± 0.05 ab
S8	83,899 ± 7918 ab	158 ± 11 bc	428 ± 55 b	424 ± 55 ab	5.2 ± 1.23 ab	0.81 ± 0.15 ab
S9	70,458 ± 3473 ab	139 ± 3 c	396 ± 33 bc	386 ± 36 b	4.6 ± 0.74 b	0.79 ± 0.11 b
S10	64,059 ± 13,164 b	149 ± 2 c	387 ± 30 bc	380 ± 32 b	5.6 ± 0.92 ab	0.89 ± 0.10 ab
S11	72,766 ± 7313 ab	165 ± 5 b	370 ± 20 bc	365 ± 20 bc	4.2 ± 0.37 b	0.74 ± 0.03 b

The different lowercase letters indicate that the difference is significant at the 0.05 level; *n* = 3.

## Data Availability

The sequence data from the coastal sediments were deposited in the Sequence Read Archive of the NCBI under accession number PRJNA833334.

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
