# Peer review of "Fungal Diversity and Its Relationship with Environmental Factors in Coastal Sediments from Guangdong, China"

_jof, 2023, doi:10.3390/jof9010101_

Round 1

Reviewer 1 Report

Dear Authors,

The manuscript titled “Fungal diversity and its relationship with environmental factors in coastal sediments from Guangdong, China” show the influence of environmental factors on marine fungal communities using metabarcoding approach. The manuscript was well-written, and appropriate methods were used in this study.

However, authors should provide more detailed descriptions for sampling methods, sampling sites and type of coastal sediments. As briefly described in discussion, each sampling sites had unique characteristics (i.e. good natural environment of Site S1). The composition of fungal microbiome could be affected by characteristics of natural habitats, as well as environmental factors (in this case, chemical properties of water and sediments). Adding brief descriptions for each sampling sites in Table 1, or grouping of sampling sites would improve the understanding of results.

Minor comments

L117: Please provide the version and date of UNITE database used in this study.

L178: Please change the label in Fig S3 (a-k to S1-S11)

L214: T. Laetisaria species name or typo?

L216 Emericellaand Didymosphaeria Emericella and Didymosphaeria. In addition, changing Emericella to its asexual state, Aspergillus would be better, following “one fungus: one name” nomenclature system.

L225 Figure5B Figure 5B

Author Response

Thank you for your review and helpful suggestions. Please refer to the attachment for the detailed reply of the author. All revised contents in text were highlighted in red in the revised manuscript.

Reviewer 2 Report

The overall study is interesting more for methodology than for results. The result significance would have been increased if you had provided more robust sampling design. Maybe you have done, but I can't see this. I think this is the major issue of this paper. Metagenomics have been gaining popularity but need rational designs before achieving enormous amounts of data. Moreover, awareness is needed when dealing with the ecological role of the detected taxa.  Also, fungal taxonomy needs revision. 

Please find the detailed comments in the attached file.

Author Response

(The authors gave the same response as above.)

Round 2

Reviewer 2 Report

Thank you for your quick revision work. Please see further requests in the attached file. Pay attention to taxonomy details and improve your discussion about FUNGuild current strength and weakness, since you have relied on this.

Author Response

Thank you again for your review and helpful suggestions. Please refer to the attachment for the detailed reply of the author. All revised contents in text were highlighted in red in the revised manuscript.
